# A Disintegrin and Metalloprotease 15 is Expressed on Rheumatoid Arthritis Synovial Tissue Endothelial Cells and may Mediate Angiogenesis

**DOI:** 10.3390/cells8010032

**Published:** 2019-01-09

**Authors:** Shinichiro Nishimi, Takeo Isozaki, Kuninobu Wakabayashi, Hiroko Takeuchi, Tsuyoshi Kasama

**Affiliations:** Division of Rheumatology, Department of Medicine, Showa University School of Medicine, Tokyo 142-8555, Japan; section.semi.express243@gmail.com (S.N.); kuninobu@med.showa-u.ac.jp (K.W.); takeuchi8223@yahoo.co.jp (H.T.); tkasama@med.showa-u.ac.jp (T.K.)

**Keywords:** ADAM15, rheumatoid arthritis, angiogenesis, cytokine

## Abstract

A disintegrin and metalloprotease 15 (ADAM15) is involved in several malignancies. In this study, we investigated the role of ADAM15 in rheumatoid arthritis (RA) angiogenesis. Soluble ADAM15 (s-ADAM15) in serum from RA and normal (NL) subjects was measured using ELISA. To determine membrane-anchored ADAM15 (ADAM15) expression in RA synovial tissues, immunohistochemistry was performed. To examine the role of ADAM15 in angiogenesis, we performed in vitro Matrigel assays and monocyte adhesion assays using human umbilical vein endothelial cells (HUVECs) transfected with ADAM15 siRNA. Finally, to investigate whether angiogenic mediators were affected by ADAM15, cytokines in ADAM15 siRNA-transfected HUVEC-conditioned medium were measured. ADAM15 was significantly higher in RA serum than in NL serum. ADAM15 was also expressed on RAST endothelial cells. ADAM15 siRNA-treated HUVECs had decreased EC tube formation in response to RA synovial fluids compared with non-treated HUVECs. The adhesion index of ADAM15 siRNA-transfected HUVECs was significantly lower than the adhesion index of control siRNA-transfected HUVECs. ENA-78/CXCL5 and ICAM-1 were decreased in tumor necrosis factor (TNF)-α-stimulated ADAM15 siRNA-transfected HUVEC-conditioned medium compared with TNF-α-stimulated control siRNA-transfected HUVEC-conditioned medium. These data show that ADAM15 plays a role in RA angiogenesis, suggesting that ADAM15 might be a potential target in inflammatory diseases such as RA.

## 1. Introduction

Rheumatoid arthritis (RA) is a common autoimmune disease characterized by the accumulation of inflammatory cells in the joints, leading to hyperproliferation of synovial cells and tissue destruction [1]. RA synovium contains high levels of inflammatory cytokines, including interleukin (IL)-6 and tumor necrosis factor (TNF)-α, which are major inflammatory mediators that induce and maintain disease processes and abundant inflammatory cells, including infiltrating lymphocytes and monocytes [2,3,4]. Anti-IL-6 and anti-TNF-α have been developed for RA and have resulted in clinical improvements during the past decade [5,6].

A disintegrin and metalloproteases (ADAMs) are membrane-anchored glycoproteins composed of multiple distinct protein modules [7]. One of these is ADAM17, also known as TNF-α-converting enzyme (TACE), which is a cell surface protease that has been implicated in the shedding of inflammatory cytokines and growth factors such as IL-6 receptor or epidermal growth factor receptor from the cell surface [8,9]. ADAM15 belongs to the family of ADAMs. ADAM15 is expressed in several solid malignant tumors, such as breast cancer and prostate cancer, and is involved in the progression to metastatic disease [10]. ADAM15 is also related to inflammatory diseases such as RA. ADAM15 is overexpressed in lining cells, endothelial cells, and macrophage-like cells in the sublining layer of RA synovium [11]. However, the role of ADAM15 in RA synovium is unclear. Here, we examine the role of ADAM15 in RA angiogenesis.

## 2. Materials and Methods

### 2.1. Patients

RA and osteoarthritis (OA) synovial fluids and sera were obtained from patients. We used data from a cohort of RA patients (2008–2012) who were diagnosed using the 1987 American College of Rheumatology (ACR) classification criteria for RA. The sera were collected from 21 patients before the initial treatment and at 12, 24, and 52 weeks after treatment with tocilizumab (TCZ). Synovial fluids (SFs) form 12 OA patients were collected. Fifty-three healthy subjects were recruited on a voluntary basis as control. RA synovial tissues (STs) were obtained from patients undergoing arthroplasty. The healthy controls were Japanese blood donors matched for gender and age. All specimens were obtained with informed consent and collected following approval from the Showa University Institutional Review Board. 

### 2.2. Cell Culture

Human umbilical vein endothelial cells (HUVECs) were purchased from Lonza (Walkersville, MD, USA) and were maintained in growth factor complete EC basal medium (Lonza). HUVECs were seeded in six-well plates at 1 × 10^5^/well. Cells were serum-starved before cytokine stimulation.

### 2.3. Enzyme-Linked Immunosorbent Assay (ELISA) of ADAM-15, CXCL16, Fractalkine/CX3CL1, ENA78/CXCL5, Intercellular Adhesion Molecule (ICAM)-1, Vascular Cell Adhesion Molecule (VCAM)-1, MCP-1/CCL2, IL-8, and TGF-β

ELISAs were performed as described previously [12]. The level of soluble ADAM15 (s-ADAM15) in serum and SFs was measured following the manufacturer’s protocol (R&D Systems, Inc., Minneapolis, MN, USA). CXCL16, fractalkine/CX3CL1, ENA78/CXCL5, intercellular adhesion molecule (ICAM), vascular endothelial growth factor (VCAM), MCP-1/CCL2, IL-8, and TGF-β in ADAM15 small interfering RNA (siRNA)-transfected- or control HUVEC-conditioned medium were measured with an ELISA kit (R&D systems).

### 2.4. RNA Extraction and Quantitative Polymerase Chain Reaction (qPCR) of HUVECs

Total RNA was isolated from HUVECs using RNeasy mini RNA isolation kits (Qiagen, Valencia, Spain) in accordance with the manufacturer’s protocol. Following isolation, the RNA was quantified and checked for purity using a spectrophotometer (Nanodrop Technologies, Wilmington, DE, USA). cDNA was prepared using a Reverse-IT MAX first-strand synthesis kit (Abgene, Rochester, NY, USA) per the manufacturer’s protocol. ADAM15 and glyceraldehyde 3-phosphate dehydrogenase (GAPDH) primer pairs were purchased from Integrated DNA Technologies (Coralville, IA, USA). ADAM15 mRNA was measured by real-time qPCR with the use of SYBR Green/ROX master mix (SABiosciences, Frederick, MD, USA) on an Mx3005P thermal cycler (Stratagene, Santa Clara, CA, USA). The ratio of each mRNA relative to the GAPDH mRNA was calculated using the ΔΔ threshold cycle method. All samples were run in duplicate and analysed using Applied Biosystems software (Life Technologies, Carlsbad, CA, USA).

### 2.5. Immunohistochemistry

RA and OA STs and TNF-α-stimulated HUVECs were fixed in ice-cold acetone for 20 min and then washed with PBS. Then, the slides were blocked with 5% goat serum and 20% FBS in PBS for 60 min at 37 °C. Rabbit IgG (10 μg/mL, Santa Cruz Biotechnology, Santa Cruz, CA, USA) or rabbit anti-human ADAM15 antibody (10 μg/mL, Abcam, Cambridge, MA, USA) was used as primary antibody. The slides were incubated at 4 °C overnight. The next day, a biotinylated goat anti-rabbit antibody (Vector, Burlingame, CA, USA) was added as a secondary antibody, and the slides were incubated for 30 min at 37 °C. The slides were incubated with the Vectastain Avidin–biotin complex (ABC) Elite regimen for 20 min. Finally, diaminobenzidine (DAB) was added for 5 min. After that, the slides were counterstained with Gill’s hematoxylin for 30 s and then washed three times with tap water. Finally, the slides were washed with 70%, 95%, 100% ethanol, and 100% isopropyl alcohol.

### 2.6. RNA Silencing

HUVECs were seeded in six-well plates at a density of 1.5 × 10^5^ cells per well. siRNAs (100 nM) against ADAM15 or control were mixed with TransIT-TKO transfection reagent (Mirus, Madison, WI, USA) according to the manufacturer’s instructions and overlaid on the cells. The cells were incubated with siRNA/TransIT-TKO for 24 h at 37 °C. ADAM15 was purchased from Santa Cruz Biotechnology. Knockdown of ADAM15 secretion was confirmed using Western blotting. The membranes were probed with rabbit anti-human ADAM15 antibody (Abcam) or anti-β-actin.

### 2.7. In Vitro HUVEC Chemotaxis Assay

Chemotaxis assays were performed using a 48-well modified Boyden chamber system. In order to confirm the role of ADAM15, we performed HMVEC chemotaxis assay using RA synovial fluids. Briefly, HUVECs were transfected with control or ADAM10 siRNA using TransIT-TKO. Control siRNA- or ADAM15 siRNA- (100 nM) treated HUVECs were placed in the bottom wells of the chambers, and diluted RA synovial fluid (1:50 in PBS) was added to the top wells of the chambers. Readings represent the number of cells migrating through the membrane (the sum of three high-power fields/well, averaged for each quadruplicate well).

### 2.8. In vitro Matrigel Tube Formation Assay

Matrigel tube formation assays using growth factor-reduced Matrigel (BD Biosciences, Bedford, MA, USA) were performed as described previously [13]. Briefly, HUVECs were transfected with control or ADAM15 siRNA using TransIT-TKO. The control used was PBS (negative control). The treated HUVECs (1.8 × 10^4^ cells) were plated on Matrigel in synovial fluid from five patients with RA (1:100) or PBS for 6 h at 37 °C. Photographs (100×) were taken, and tubes were counted by a blinded observer. Tubes were defined as elongated connecting branches between two identifiable HUVECs.

### 2.9. Monocyte Adhesion Assay

The adhesion of THP-1 cells to control siRNA-treated or ADAM15 siRNA-treated HUVECs grown to confluence in 96-well plates was examined. HUVECs were serum-starved overnight. The next day, the cells were treated with TNF-α (10 ng/mL) for 24 h. THP-1 cells were collected and labeled with Calcein AM fluorescent dye (Life Technologies, 5 μM) for 30 min. After being washed twice, 1 × 10^5^ THP-1 cells were added to each well and incubated for 30 min at room temperature. Nonadherent cells were washed away, and the fluorescence was measured using a Synergy HT fluorescence plate reader (BioTek Instruments, Winuski, VT, USA).

### 2.10. Statistical Analysis

Data were analyzed using the *t*-test, assuming equal variances. Data are reported as means ± SEM. *p* values lower than 0.05 were considered significant.

## 3. Results

### 3.1. ADAM15 is Expressed in RA Sera and Synovial Fluids

We investigated the levels of s-ADAM15 in sera and SFs. The base line characteristics of the study group are shown in Table 1. We measured the levels of s-ADAM15 in RA and NL sera using ELISA. The level of s-ADAM15 was higher in RA sera than in NL sera (509 ± 20 pg/mL and 314 ± 44 pg/mL, respectively, *p* < 0.05, Figure 1A). We next measured the levels of s-ADAM15 in RA and OA synovial fluids. The levels of s-ADAM15 were higher in RA synovial fluids than in OA synovial fluids (619 ± 53 pg/mL and 328 ± 39 pg/mL, respectively, *p* < 0.05, Figure 1B). Furthermore, we examined the levels of s-ADAM15 in RA patients who were treated with tocilizumab at 12, 24, and 54 weeks after treatment. s-ADAM15 was significantly decreased in the sera at 24 and 54 weeks compared with the level before treatment (432 ± 21 pg/mL, 434 ± 22 pg/mL and 500 ± 21 pg/mL, respectively, *p* < 0.05, Figure 1C). 

### 3.2. ADAM15 is Expressed in RA ST Endothelial Cells (ECs)

To examine the expression of ADAM15 in RA ST, immunohistochemistry was performed. We found that ADAM15 was expressed in RA ST ECs (Figure 2A–C). On the other hand, ADAM15 was not expressed in OA ST ECs (Figure 2D). To determine which mediators were involved in ADAM15 expression in ECs, HUVECs were incubated with TNF-α. ADAM15 was significantly elevated in TNF-α-stimulated HUVEC-conditioned medium compared with non-stimulated HUVEC-conditioned medium (Figure 2E). On the other hand, ADAM15 mRNA was not increased by TNF-α, IL-1β, IL-6, or interferon-γ stimulation (Figure 2F).

### 3.3. ADAM15 Mediates the Release of Potent Angiogenic Factors

To confirm the function of ADAM15 in HUVECs, we used siRNA directed against ADAM15. The knockdown of ADAM15 in HUVECs was confirmed by Western blotting, showing that ADAM15 expression was lower than in control cells (Figure 3A). To determine the role of ADAM15 in angiogenesis, we performed in vitro chemotaxis assays and Matrigel tube formation assays. RA synovial fluid is rich in angiogenic mediators. Therefore, we tried to determine the role of ADAM15 in RA angiogenesis. We found that ADAM15 siRNA-treated HUVECs showed decreased migration compared with control siRNA-treated cells towards RA synovial fluids (4.1 ± 0.7 and 1.9 ± 0.1, *p* < 0.05, Figure 3B). Hence, we demonstrated ADAM15 siRNA-treated HUVECs had decreased EC tube formation in Matrigel in response to RA SFs compared with non-treated HUVECs (number of EC tubes 4 ± 1 and 1 ± 0, respectively, *p* < 0.05, Figure 3C–E). Finally, to further examine the role of ADAM15, we performed in vitro adhesion assays. The adhesion index of ADAM15 siRNA-transfected HUVECs was significantly lower than the adhesion index of control siRNA-transfected HUVECs (9435 ± 1085 and 5659 ± 1084, respectively, *p* < 0.05, Figure 4). In addition, the adhesion index of TNF-α-stimulated HUVECs was significantly lower than the adhesion index of control siRNA-transfected HUVECs (3539 ± 622.8 and 1914 ± 281.4, respectively, *p* < 0.05, Figure 4).

To determine whether ADAM15 cleaves angiogenic factors from HUVEC surfaces, ELISA was performed. ENA78/CXCL5 was significantly decreased in TNF-α-stimulated ADAM15 siRNA-transfected HUVEC-conditioned medium compared with TNF-α stimulated control HUVEC-conditioned medium (83 ± 5 pg/mL, 0 ± 0 pg/mL, *p* < 0.05, Figure 5A). ICAM-1 was also significantly decreased in TNF-α-stimulated ADAM15 siRNA-transfected HUVEC-conditioned medium compared with TNF-α-stimulated control HUVEC-conditioned medium (1845 ± 11 pg/mL, 164 ± 3 pg/mL, *p* < 0.05 Figure 5B). On the other hand, CXCL16, fractalkine/CX3CL1, VCAM-1, MCP-1/CCL2, IL-8, and TGF-β were not decreased in TNF-α-stimulated ADAM15 siRNA-transfected HUVEC-conditioned medium compared with TNF-α-stimulated, non-transfected HUVEC-conditioned medium.

## 4. Discussion

ADAM15 is one of the ADAM family members and is related to the shedding of proteins from the cell surface [14]. In this study, we first found that ADAM15 was expressed in RA synovium and serum and was decreased after treatment. ADAM15 expression is reported in several solid malignant tumors, such as breast and prostate cancers [10,15,16]. Kuefer et al. suggested that ADAM15 plays a role in the progression to metastatic disease [10]. Regarding autoimmune diseases, Mosnier et al. reported that ADAM15 is expressed in inflammatory bowel disease colon epithelium and endothelial cells [17]. Gao et al. also reported high levels of ADAM15 mRNA in fibroblasts [18]. Our results support their findings. We found that the levels of ADAM15 were higher in RA sera compared with NL sera. ADAM15 was also significantly higher in RA SFs compared with OA SFs. We first found that ADAM15 was decreased after treatment with tocilizumab in patients with RA. Taken together, our findings indicate that ADAM15 could be involved in RA inflammation.

We have also shown that ADAM15 was expressed on RA ST ECs and HUVECs. Komiya et al. demonstrated that RA tissue ECs were responsible for the expression of ADAM15 [11]. Bohm et al. also found that ADAM15 was overexpressed in RA ST compared with normal tissue [19]. Hence, Sun et al. showed that ADAM15 regulated endothelial permeability, which is considered one of the key cellular processes in the development of inflammatory disorders [20]. These findings indicate that ECs are important sources of ADAM15 expression. In addition, we found ADAM15 mRNA in HUVECs was not increased by TNF, IL-1, IL-6, or interferon-gamma stimulation. These results indicate soluble ADAM15 is involved in RA disease activity. On the other hand, membrane ADAM15 may be the cleaving mediator. Therefore, we next focused on the role of ADAM15 in ECs, especially angiogenesis.

Angiogenesis plays an important role in the pathogenesis of RA [21]. To determine the role of ADAM15 in angiogenesis, we performed in vitro chemotaxis assays and Matrigel tube formation assays. EC chemotaxis is an initial step in the angiogenic process, and the RA joint is rich in angiogenic mediators. Therefore, we performed EC chemotaxis towards RA synovial fluids. We found ADAM15 siRNA-treated HUVECs showed decreased migration compared with control siRNA-treated cells. We next performed in vitro Matrigel tube formation assays and adhesion assays. We found that ADAM15 siRNA-treated HUVECs had decreased EC tube formation in Matrigel in response to RA SFs compared with non-treated HUVECs, and the adhesion index of transfected HUVECs was significantly lower than that of control siRNA-transfected HUVECs. Tube formation is the first step in angiogenesis. The limitation of this study is that it did not directly assess angiogenesis. We have to perform angiogenesis assays in the future. Neali et al. showed that high levels of ADAM15 expression suggest that it may play a role in endothelial function or in tumors [16]. Chatchawit et al. demonstrated by using genome-wide RA data that ADAM15 may contribute to abnormal angiogenesis in RA [22]. In this study, we demonstrated that ADAM15 siRNA-treated HUVECs had decreased EC line and tube formation. Moreover, the adhesion index of ADAM15 siRNA-transfected HUVECs was significantly decreased compared with that of control siRNA-transfected HUVECs. ENA-78/CXCL5 is a chemokine that is involved in neutrophil invasion and angiogenesis and is present in RA synovial fluid and synovial tissue [23]. ICAM-1 is associated with the adhesion of activated lymphocytes to ECs as one of the adhesion molecules controlling biotaxis. In addition, ICAM-1 is also expressed in inflammatory arthritis [24]. Here, we have shown that ENA78/CXCL5 and ICAM-1 were significantly decreased in TNF-α-stimulated ADAM15 siRNA-transfected HUVEC-conditioned medium compared with TNF-α-stimulated control medium, suggesting that ADAM15 plays an important role in angiogenesis in RA through regulating the expression of ENA-78/CXCL5 and ICAM-1. We confidently think that TNF-α is important in rheumatoid angiogenesis. In this study, we investigated if ADAM15 is involved in rheumatoid angiogenesis. ADAM15 was increased by TNF-α stimulation. On the other hand, ADAM15 did not shed TNF-α. Taken together, TNF-α is upstream of ADAM15 in rheumatoid angiogenesis, and ADAM15 is directly involved in angiogenesis via unidentified mediators.

As per the previous paragraph, ADAM15 is directly involved in rheumatoid angiogenesis via the production of proinflammatory mediators such as ENA78/CXCL5 and ICAM-1.

## 5. Conclusions

We report here that ADAM15 is overexpressed in RA sera, SFs, and STs. In addition, ADAM15 level in RA sera was decreased using TCZ therapy. These results indicate that the ADAM15 level in serum correlated with RA disease activity. We also demonstrated that inhibition of ADAM15 in ECs resulted in reduced EC tube formation and monocyte adhesion. Overall, our results indicate that ADAM15 plays a seminal role in the pathogenesis of inflammatory arthritis, suggesting that ADAM15 may be a therapeutic target in RA.

## Figures and Tables

**Figure 1 cells-08-00032-f001:**
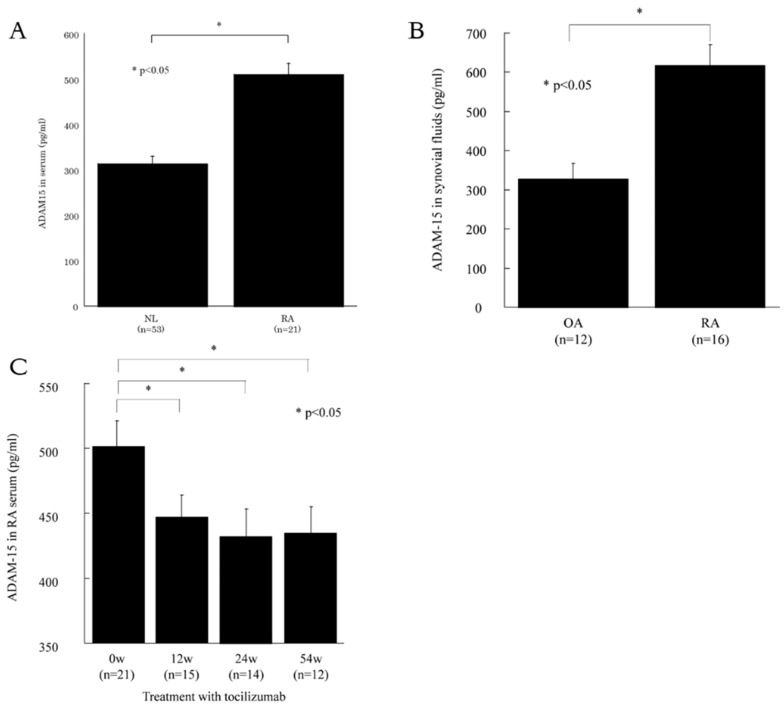
ADAM 15 is expressed in sera and synovial fluids (SFs). (**A**) Levels of ADAM15 were higher in RA sera than in normal (NL) sera. (**B**) The levels of ADAM15 were higher in RA SFs than in osteoarthritis (OA) SFs. (**C**) Levels of ADAM15 in patients who were treated with tocilizumab. ADAM15 was significantly decreased 24 and 54 weeks after tocilizumab treatment compared with the level prior to treatment. * *p* < 0.05 was significant.

**Figure 2 cells-08-00032-f002:**
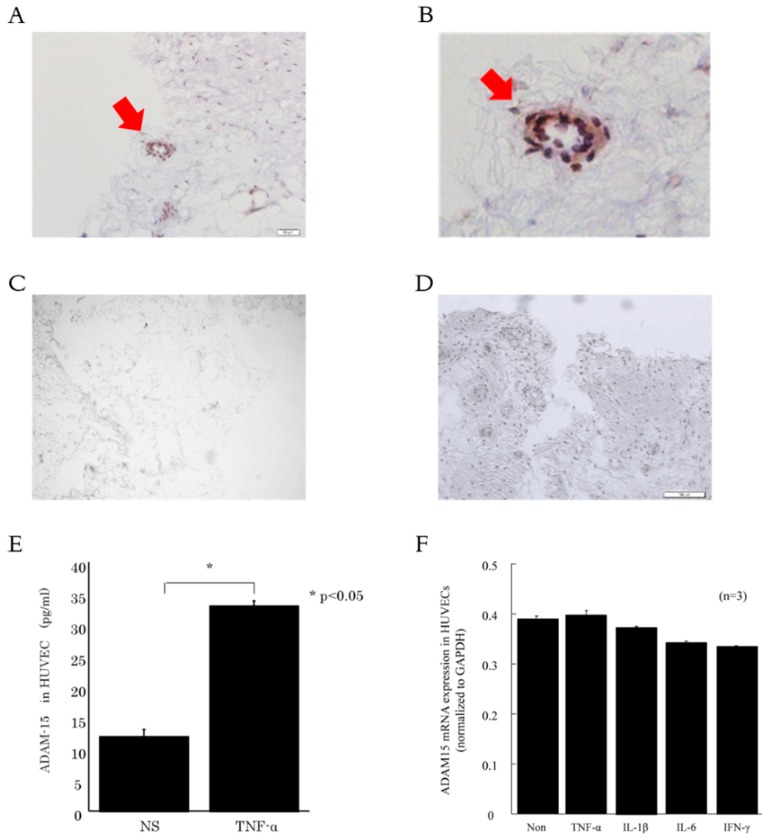
ADAM15 is expressed in RA synovial tissue (ST) endothelial cells (ECs). (**A** and **B**) Representative photomicrographs of STs from RA patients. (**C**) Representative photomicrographs of STs from OA patients. STs were stained for ADAM15. Cryosections and cultured cells were stained for ADAM15 (**A**, **B,** and **D**) or control IgG (**C**). (**E**) ADAM15 was increased in TNF-α-stimulated human umbilical vein endothelial cells (HUVEC)-conditioned medium compared with non-stimulated HUVEC-conditioned medium. (**F**) ADAM15 mRNA level was not increased by TNF, IL-1, IL-6, or interferon (IFN)-gamma stimulation (*n* = 3). * *p* < 0.05 was significant.

**Figure 3 cells-08-00032-f003:**
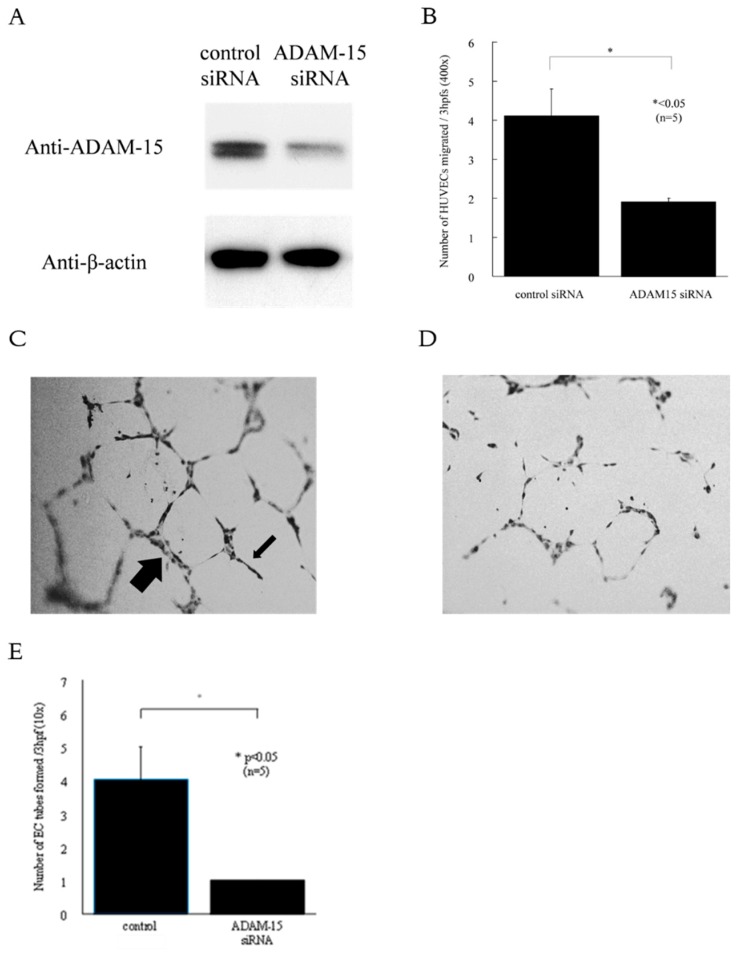
ADAM15 siRNA inhibits EC chemotaxis and EC tube formation. (**A**) Knockdown of ADAM15 was confirmed by Western blotting. (**B**) ADAM15 contributes to RA synovial fluid-mediated HUVEC chemotaxis. RA synovial fluid was depleted of rheumatoid factor. HUVECs were treated with control or ADAM10 siRNA. (*n* = number of different RA synovial fluid samples used). (**C** and **D**) RA synovial fluids were used to stimulate HUVEC tube formation on Matrigel. Photographs were taken at 100×. Representative photographs of non-treated HUVECs (**C**) and ADAM15 siRNA-treated HUVECs (**D**). RA synovial fluid-stimulated HUVECs treated with ADAM15 siRNA formed significantly fewer tubes on Matrigel than non-treated HUVECs (*n* = number of patients). For both chemotaxis and tube formed assays, means are presented with SEM, * *p* < 0.05 was significant; (hpfs, high-power fields.).

**Figure 4 cells-08-00032-f004:**
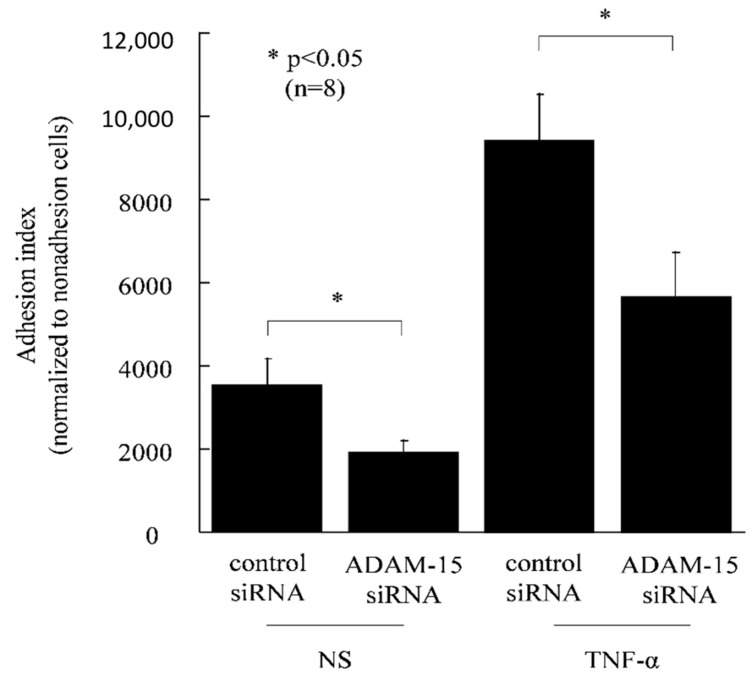
ADAM15 siRNA reduces monocyte adhesion. The adhesion index of ADAM15 siRNA-transfected HUVECs was significantly lower than the adhesion index of control siRNA-transfected HUVECs. * *p* < 0.05 was significant.

**Figure 5 cells-08-00032-f005:**
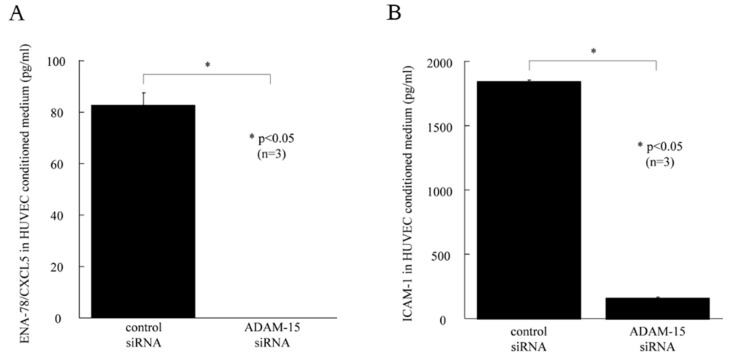
ADAM15 siRNA reduces ENA78/CXCL5 and ICAM-1 expression. (**A**) ICAM-1 was significantly decreased in TNF-α-stimulated ADAM15 siRNA-transfected HUVEC-conditioned medium compared with TNF-stimulated control medium. (**B**) ICAM-1 was significantly decreased in TNF-α-stimulated ADAM15 siRNA-transfected HUVEC-conditioned medium compared with TNF-α-stimulated control medium. * *p* < 0.05 was significant.

**Table 1 cells-08-00032-t001:** Characteristics of the study population. RA: rheumatoid arthritis.

Total (n)	21
Gender (female:male)	20:1
Mean age (years)	51.7 ± 3.43
Duration of RA (years)	7.3 ± 1.9
Baseline DAS28 (ESR)	4.83 ± 0.29
Dosage of MTX (mg)	6.48 ± 1.16
Dosage of prednisolone (mg)	3.64 ± 0.56

ESR: erythrocyte sedimentation rate; MTX: methotrexate.

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
