# Peer review of "A Disintegrin and Metalloprotease 15 is Expressed on Rheumatoid Arthritis Synovial Tissue Endothelial Cells and may Mediate Angiogenesis"

_cells, 2019, doi:10.3390/cells8010032_

Round 1

Reviewer 1 Report

Materials and methods section – the Authors should add information about number of OA patients recruit for this study.

The Authors should describe healthy subjects. For example – geographic region, if they were blood donors or not.

Please add the Table which will be presented the baseline characteristics of study groups. What was DAS-28 before treatment?

Why only 5 healthy subjects were recruit for this study? This group is too small for comparison.

Author Response

Materials and methods section – the Authors should add information about number of OA patients recruit for this study.

We collected 12 OA patient’s synovial fluid. We have added this information in Materials and Methods section on page 4.

The Authors should describe healthy subjects. For example – geographic region, if they were blood donors or not.

This healthy controls were matched with gender and age in Japanese blood donors. We added this information about healthy subjects in Materials and Methods section on page 4.

Please add the Table which will be presented the baseline characteristics of study groups. What was DAS-28 before treatment?

We showed patient characteristics in Table 1. DAS28 before treatment was 4.83 ± 0.29. We added the sentence [We showed background of RA patients in Table 1.] in Result on page 9.

Why only 5 healthy subjects were recruit for this study? This group is too small for comparison.

We measured ADAM15 in 53 healthy controls. The data was shown in Figure 1.

Reviewer 2 Report

cells-394382

A disintegrin and metalloprotease 15 is expressed on rheumatoid arthritis synovial tissue endothelial cells and mediates angiogenesis

Nishimiet al.

This manuscript addressed the potential role of ADAM15 in synovial tissue endothelial cells and angiogenesis.  ADAM15 is one of the ADAMs whose roles are not well defined. Thus this paper’s intention to address this is very good. However, the study is very phenomenological and assay employed is not well-defined one. As it is data does not support conclusions made.

Specific points

1. Authors mixed soluble ADAM15 and membrane-anchored ADAM15. The level of ADAM15 in the serum or synovial fluid is soluble ADAM15, and it is not clear if the expression of ADAM15 is indeed increased or not. 

2.  In Fig 2, immunohistochemistry data is likely indicating membrane-bound ADAM15. The TNF-induced increase of ADAM15 in HUVEC media is a soluble ADAM15. This data can be interpreted as the increased shedding of ADAM15 upon TNF treatment as well. Authors need to address mRNA level is also increased or not upon TNF treatment. Since the role of ADAM15 as a soluble form is likely to be different from membrane-bound form, authors need to define what is happening here.

3. Why the did authors examine only TNF as a stimulus? What about IL-1, IL-6, IL-10, IL-23, IFN gamma?  

4. In Figure 3, authors addressed the role of ADAM15 by knocking down ADAM15 in HUVEC (70% reduction). Panel B is showing classical Matrigel cord formation assay which does not reflect the ability of HUVEC to for tubular structure.  This is not an angiogenesis assay. Authors need to adopt proper angiogenesis assay. This can be said for Fig 4 as well.

5. Figure 5 is interesting.  But the data is far from completion. Authors need to analyze mRNA level. Also adding conditioned media that contains soluble ADAM15 may need to investigate. Is it a soluble form that is the necessary or membrane-bound form?

Overall, the study is premature, and authors consider to make the study comprehensive.  

Author Response

Authors mixed soluble ADAM15 and membrane-anchored ADAM15. The level of ADAM15 in the serum or synovial fluid is soluble ADAM15, and it is not clear if the expression of ADAM15 is indeed increased or not. 

Thank you for your suggestion. We have changed that soluble ADAM-15 is s-ADAM15 and membrane-anchored ADAM15 is ADAM15.

In Fig 2, immunohistochemistry data is likely indicating membrane-bound ADAM15. The TNF-induced increase of ADAM15 in HUVEC media is a soluble ADAM15. This data can be interpreted as the increased shedding of ADAM15 upon TNF treatment as well. Authors need to address mRNA level is also increased or not upon TNF treatment. Since the role of ADAM15 as a soluble form is likely to be different from membrane-bound form, authors need to define what is happening here.

We measured ADAM15 mRNA with TNF treatment. ADAM15 mRNA was not increased with TNF. We showed soluble ADAM15 is increased with TNF stimulation. ADAM15 is released from HUVECs. Taken together, soluble ADAM15 is involved in RA disease activity. On the other hand, membrane ADAM15 may be cleaving mediators. We added this thought in discussion section on page 12 and 13.

Why did authors examine only TNF as a stimulus? What about IL-1, IL-6, IL-10, IL-23, IFN gamma? 

We measured ADAM15 mRNA in TNF, IL-1, IL-6 and IFN-gamma. Soluble ADAM15 in TNF stimulated HUVEC conditioned medium was increased compared with nonstimulated HUVEC conditioned medium. However, ADAM15 mRNA was not increased after TNF, IL-1, IL-6 or IFN gamma stimulation. We added this data in results section on page 10.

In Figure 3, authors addressed the role of ADAM15 by knocking down ADAM15 in HUVEC (70% reduction). Panel B is showing classical Matrigel cord formation assay which does not reflect the ability of HUVEC to for tubular structure.  This is not an angiogenesis assay. Authors need to adopt proper angiogenesis assay. This can be said for Fig 4 as well.

We totally understand this point. Tube formation assay is first step in angiogenesis. We have to address angiogenesis assay. Therefore, we mention this point in results section on page 13.

Figure 5 is interesting.  But the data is far from completion. Authors need to analyze mRNA level. Also adding conditioned media that contains soluble ADAM15 may need to investigate. Is it a soluble form that is the necessary or membrane-bound form?

We mentioned #2 and #3. ADAM15 mRNA was not increased with TNF treatment.

Reviewer 3 Report

The authors demonstrated that ADAM15 plays a role in RA angiogenesis and suggested that the molecule might become a potential target in RA treatment. Some findings are quite interesting. However, a number of comments and queries are raised below for authors’ references.

M &M 2.2 Cell Culture: a typing error of vascular endothelial adhesion molecule, but not growth factor, is the abbreviation of VCAM. In addition, the authors have measured the production of MCP-1/CCL2, IL-8, and TGF-b by ELISA. But the results did not appear in the text, figures or tables.

It conceivable that TNF-a can mediate angiogenesis, adhesion molecule expression on endothelial cells, binding capacity of mononuclear cells with EC, and different proinflammatory cytokines/chemokines/growth factors expression of immune-related cells. These effects are quite similar to ADAM15. The authors should clarify which molecule, TNF-a or ADAM15, is the up-stream for rheumatoid angiogenesis?

The authors argue that ADAM15 play an important role in rheumatoid angiogenesis. The authors should explain why increasing ADAM15 expression in rheumatoid joint in the “Discussion” section.

Which cytokines/chemokines/growth factors in rheumatoid joint involve in enhanced ADAM15 expression?

Author Response

1.           M &M 2.2 Cell Culture: a typing error of vascular endothelial adhesion molecule, but not growth factor, is the abbreviation of VCAM. In addition, the authors have measured the production of MCP-1/CCL2, IL-8, and TGF-b by ELISA. But the results did not appear in the text, figures or tables.

Thank you for your pointing out. We have corrected spelling. We also added the date of CXCL16, fractalkine/CX3CL1, VCAM-1, MCP-1/CCL2, IL-8, and TGF-β in results section on page 11.

2.           It conceivable that TNF-a can mediate angiogenesis, adhesion molecule expression on endothelial cells, binding capacity of mononuclear cells with EC, and different proinflammatory cytokines/chemokines/growth factors expression of immune-related cells. These effects are quite similar to ADAM15. The authors should clarify which molecule, TNF-a or ADAM15, is the up-stream for rheumatoid angiogenesis?

Thank you for your suggestion. We confidently think that TNFconcern in rheumatoid angiogenesis. In this study, we would like to investigate that ADAM15 is concern in rheumatoid angiogenesis. ADAM15 was increased with TNF-α stimulation. On the other hand, ADAM15 was not shedding TNF-α. Taken together, TNF-αis the up-stream for rheumatoid angiogenesis and ADAM15 is directly involved in angiogenesis via some mediator production. We addressed this thought in discussion section on page 14.

3.           The authors argue that ADAM15 play an important role in rheumatoid angiogenesis. The authors should explain why increasing ADAM15 expression in rheumatoid joint in the “Discussion” section.

As per the previous paragraph, ADAM15 is directly involved in rheumatoid angiogenesis via proinflammatory mediator production such as ENA78/CXCL5 and ICAM-1. We mentioned it in Discussion section on page 14.

4.           Which cytokines/chemokines/growth factors in rheumatoid joint involve in enhanced ADAM15 expression?

We measured ADAM15 mRNA with TNF, IL-1, IL-6 and IFN gamma. ADAM15 was not increased with these cytokine stimulation.

Round 2

Reviewer 2 Report

This a revised manuscript. However, the authors did not address my major concern adequately. The major problem still remained in this manuscript is that none of the data support the conclusion that ADAM15 mediates angiogenesis. As I have previously stated, the matrigel cord assay is not an angiogenesis assay. I do recognize that the study has some interesting observations, but one cannot conclude that ADAM15 mediates angiogenesis assay with the data provided. Authors are urged to add proper angiogenesis assay data.

English still needs improvement too.

Author Response

Thank you for your advice. EC chemotaxis is an initial step in the angiogenic process, and the RA joint is rich in angiogenic mediators. Therefore, we performed EC chemotaxis towards RA synovial fluids. We found that ADAM15 siRNA treated HUVECs showed decreased migration compared with control siRNA, towards RA synovial fluids (4.1 ± 0.7 and 1.9 ± 0.1, p<0.05).

We added this new angiogenesis assay data in results and discussion sections on page 10 and 13.